# A PHYSICS-INFORMED NEURAL NETWORK FOR COUPLED CALCIUM DYNAMICS IN A CABLE NEURON

**Zachary M. Miksis & Gillian Queisser**
Department of Mathematics
Temple University
Philadelphia, PA 19122
`{miksis,queisser}@temple.edu`

## ABSTRACT

Transcranial magnetic stimulation (TMS) is a noninvasive treatment for a variety of neurological and neuropsychiatric disorders by triggering a calcium response through magnetic stimulation. To understand the full effects of this treatment, researchers will often use numerical simulations to model and study the calcium response. These simulations are limited to short-time simulations of single neurons due to computational complexity, restricting their use in clinical settings. In this paper, we explore an application of physics-informed neural networks (PINNs) to accurately produce long-time simulations of neuronal responses, opening the possibility of utilizing these methods in clinical applications to directly benefit patients.

## 1 INTRODUCTION

Transcranial magnetic stimulation (TMS) is an invaluable tool for treating a variety of neurological and neuropsychiatric disorders in a noninvasive way by using a time-varying magnetic field passed through the brain to stimulate neurons (Barker et al., 1985; Hallett, 2007). This stimulation triggers a response from intracellular calcium, which is vital to regulating the transfer of information from synaptic sites to the cell nucleus (Borole et al., 2023). Ultimately, this allows for treatment of various forms of degenerative neurological disorders, and TMS is therefore used extensively in both research and clinical settings (Lefaucheur et al., 2014).

While TMS is very important, its full effects are still not entirely understood. To investigate these effects, computational tools and studies are exceptionally important to complement experimental studies (Grein et al., 2014). To this end, different numerical computation tools have been developed (Borole, 2022; Shirinpour et al., 2021; Guan & Queisser, 2022). While these tools are very good at capturing fine grain details and the quick responses to fast TMS frequencies, they are limited to short simulated time spans of single neurons due to computational complexity. In order to address these limitations, we look to neural networks (from here on, "neuron" will be used to refer to a biological neuron, and "neural network" will be used to refer to the machine learning paradigm).

Neural networks have been used in various applications, many focusing on image recognition and reconstruction (e.g., see Abraham et al. (2023)). More recently, knowledge of physical models have been introduced into neural networks to form physics-informed neural networks (PINNs) (Raissi et al., 2019; 2017a;b). In this work, we utilize these PINNs to incorporate partial differential equations (PDEs) that model calcium dynamics in neurons into the larger neural network. With a network that satisfies physical laws, we can build simulations that provide long-time results, overcoming one of the limitations of traditional numerical methods. This leads to another limitation of PINNs, in that they have been shown to have difficulty accurately to simulate diffusion problems (Saadat et al., 2022). This can be overcome with strategies of both relaxation of the loss function and periodic activation functions (Snyder et al., 2023; Sitzmann et al., 2020), which we utilize in our application.

This paper is structured as follows: In Section 2, we describe the physical model of calcium dynamics that is used (with further details in Appendix A and B). In Section 3, we describe the details on PINNs as we used them in our application. In Section 4, we present results of a simulated cable neuron, with

additional plots in Appendix C. In Section 5, we draw conclusions from our results and describe the broader impacts of this work.

## 2 CALCIUM DYNAMICS

Within a neuron, ion dynamics may be modeled by general 1D reaction-diffusion equations of the form

$$\frac{\partial u}{\partial t} = D\Delta u + R(u), \tag{1}$$

where $u$ is an ion concentration, $D$ is the diffusion coefficient, and $R(u)$ is a reaction term. From Fick's first law, we have

$$J = -D\frac{\partial u}{\partial x}, \tag{2}$$

where $J$ is the diffusion flux of the ion. This leads to the Neumann boundary condition

$$\frac{\partial u}{\partial x} = -\frac{J}{D}. \tag{3}$$

In the cytosol, the non-organelle interior of a neuron, calcium is transported via various mechanisms and is buffered by the molecule calbindin. We model this using a dimension-reduced system (Borole et al., 2023), given by

$$\frac{\partial c_c}{\partial t} = \nabla \cdot (D_c \nabla c_c) + f(b, c_c) + J_{PM}, \tag{4}$$

$$\frac{\partial b}{\partial t} = \nabla \cdot (D_b \nabla b) + f(b, c_c), \tag{5}$$

where the reaction term

$$f(b, c_c) = k_b^- (b^{tot} - b) - k_b^+ b c_c$$

models the reaction equation

$$\text{Ca}^{2+} + \text{CalB} \underset{\kappa_b^+}{\overset{\kappa_b^-}{\rightleftharpoons}} \text{CalBCa}^{2+}, \tag{6}$$

and $J_{PM}$ represents the net $\text{Ca}^{2+}$ ion flux across the plasma membrane (separating intra- and extracellular space). This term can be broken down into the different mechanisms that transport calcium across the plasma membrane,

$$J_{PM} = -J_P - J_N + J_{SYN} + J_{VDCC}, \tag{7}$$

where $J_P$ is the flux from plasma membrane $\text{Ca}^{2+}$-ATPase pumps (PMCA), $J_N$ is the flux from $\text{Na}^+/\text{Ca}^{2+}$ exchangers (NCX), $J_{SYN}$ is the flux through the post-synaptic density (PSD), and $J_{VDCC}$ is the flux from voltage dependent calcium channels (VDCCs). Details of how each flux term is calculated are given in Appendix A.

## 3 PHYSICS-INFORMED NEURAL NETWORKS

To simulate calcium dynamcis, we incorporate the above diffusion model into a physics-informed neural network (PINN) (Raissi et al., 2019; 2017a;b). Consider a general partial differential equation of the form

$$u_t + \mathcal{N}[u] = 0, \tag{8}$$

with $x \in \Omega$, $t \in [0, T]$, where $u(x, t)$ is the latent solution, $\mathcal{N}[\cdot]$ is a nonlinear differential operator, and $\Omega$ is the computational domain. Define $f(x, t)$ as the left-hand-side of Eq. (8),

$$f := u_t + \mathcal{N}[u]. \tag{9}$$

By approximating $u(x, t)$ using a deep neural network, we build a physics-informed neural network $f(x, t)$ by combining this network with Eq. (9). The shared terms of the networks are learned by minimizing the mean squared error loss

$$MSE = (1 - \alpha)MSE_u + \alpha(MSE_f + MSE_b), \tag{10}$$

where

$$MSE_u = \frac{1}{N_u} \sum_{i=1}^{N_u} |u(t_u^i, x_u^i) - u^i|^2 \tag{11}$$

represents the error of the initial condition,

$$MSE_f = \frac{1}{N_f} \sum_{i=1}^{N_f} |f(t_f^i, x_f^i)|^2 \tag{12}$$

represents the error of the operator $f$ inside the domain, and

$$MSE_b = \frac{1}{N_b} \sum_{i=1}^{N_b} |u_x(t_b^i, x_b^i) + J/D|^2 \tag{13}$$

represents the error of the Neumann boundary conditions. Here, $\{t_u^i, x_u^i, u^i\}_{i=1}^{N_u}$ represents the initial condition training data, $\{t_f^i, x_f^i\}_{i=1}^{N_f}$ represents the collocation points within the domain (excluding the boundaries), and $\{t_b^i, x_b^i\}_{i=1}^{N_b}$ represents the boundary collocation points. In our minimization, we apply a relaxation parameter $\alpha \in (0, 1]$. This is selected *a priori* by trial and error, and may require tuning to fit a particular model (Snyder et al., 2023).

As demonstrated in Saadat et al. (2022) and Sitzmann et al. (2020), diffusion problems such as the calcium model are partiularly difficult to model accurately with neural networks. PINNs were initally introduced with `tanh` activation function, and were demonstrated to perform well for advection-dominated problems (Raissi et al., 2019). The *SIREN* network introduced the use of periodic activation functions (Sitzmann et al., 2020), which have been demonstrated to correct for problems in modeling diffusion-dominated problems (Saadat et al., 2022). Therefore, based on the results of the comparison of `sin` and `tanh` activation functions for presented in (Saadat et al., 2022), we utilize a `sin` activation function in our network to provide consistent long-time modeling.

## 4  RESULTS

In this section, we demonstrate the effectiveness of PINNs in modeling calcium dynamics. All constants used are given in Appendix B. We consider the computational domain $(x, t) : [-1, 1] \times [0, 1]$, and use a numerically computed solution as the exact solution (see Borole (2022) for details on numerical methods). The network consists of 4 hidden layers of 50 computational neurons each. The network is initialized with a uniform *Xavier* scheme, and an *Adam* optimizer with learning rate of 0.001 is used to minimize the $MSE$. $N = 150$ collocation points are randomly generated on both the interior of the domain and the boundary (50 on each spatial boundary and 50 on the initial condition). These can be seen in Fig. 1.

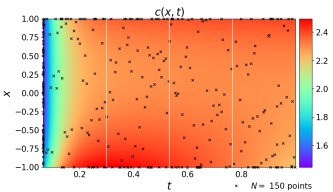

Figure 1: Domain collocation points

The full set of data points used is the discretized $(x, t) : [-1, 1] \times [0, 1]$ domain consisting of 101 spatial gridpoints and 7,193 temporal steps (for a total of 726,493 spatiotemporal data points). Of these points, the training set only consists of the 50 known initial condition ($t = 0$) points (defined by Eq. (11)), the validation set consists of the 50 points on each spatiotemporal boundary and 150 points on the domain interior that must satisfy the PDE (for a total of 250 data points, defined by Eqs. (12)-(13)), and the remaining data points makeup the test set.

It is important to note that while this case seems like a fairly simple problem that may be solved without PINN, this is only because this test case has been chosen to be small in early stages of this work. In a realistic application of simulating a full network of physical neurons over a full TMS treatment regime, a traditional numerical method will require tens of millions of timesteps over hundreds of thousands of spatial nodes. Even though this is a one-dimensional PDE, this quickly becomes an intractable simulation to run in any reasonable amount of time for use in a clinical setting. A pretrained PINN may be able to provide simulated results without significant computing resources, and can move this work beyond well-funded research and into a clinical setting.

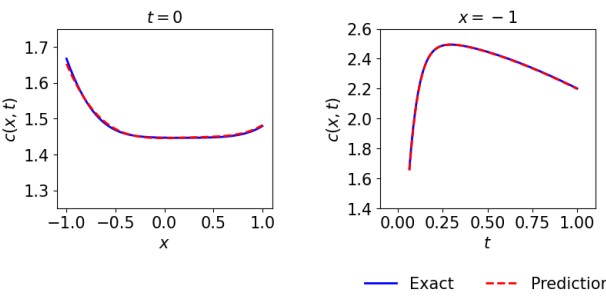

Figure 2: Calcium concentration at $t = 0$ and $x = -1$

For the biological model, we introduce a constant stimulus at the soma for the entire simulation time, and a constant voltage is applied. As can be seen in Appendix A, the gating function for VDCCs requires the solution of two ODEs. For the purposes of simulating TMS treatment protocols, predetermined voltage data may be provided to review its effects. Since the gating function only requires voltage and temporal values, these gating probabilities may be precomputed prior to training the network. We therefore avoid the requirement of modeling additional ODEs, and provide these gating probabilities explicitly to the network.

We first look at the values of the initial calcium and calbindin concentrations. In the numerical simulation, the initial concentrations are given as constant values. These may not represent the true steady state, and they will not satisfy the diffusion model, which leads to error in the PINN if they are used. We therefore consider the initial condition as the computed values after a small number of numerical timesteps, as the concentrations are approaching values that satisfy the diffusion model. As the number of numerical timesteps is selected *a priori*, the values are still not guaranteed to be consistent with the diffusion model, but this significantly improves the performance of the PINN.

In Fig. 2 and Fig. 3, we show the results of the network prediction of the initial concentrations. While there is a small amount of error, we can see that the network fairly accurately predicts these values. In Fig. 4 and Fig. 5, we see that this error is significantly reduced to nearly vanishing as the simulation moves forward in time and the concentration values become consistent with the diffusion model.

For TMS treatment purposes, we may be interested in what is occurring at the soma (cell body) of the neuron. Fig 2 and Fig 3 show the concentration over time at the soma of our cable

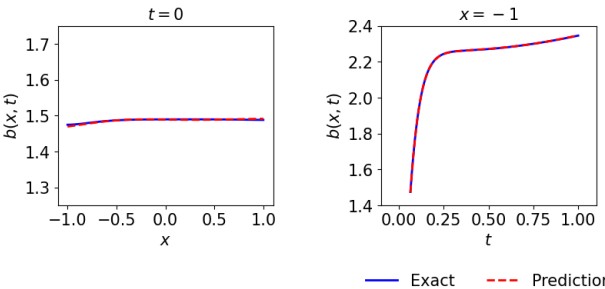

Figure 3: Calbindin concentration at $t = 0$ and $x = -1$

neuron. We see that the network very accurately predicts the concentrations of both calcium and calbindin over time, producing a solution nearly indistinguishable from the exact solution.

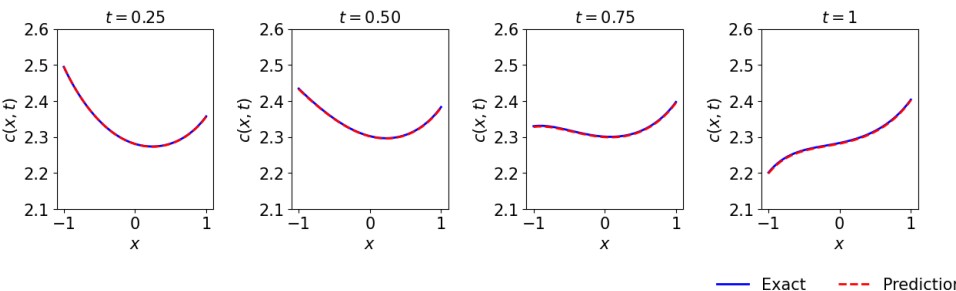

Figure 4: Calcium concentration along neuron over time

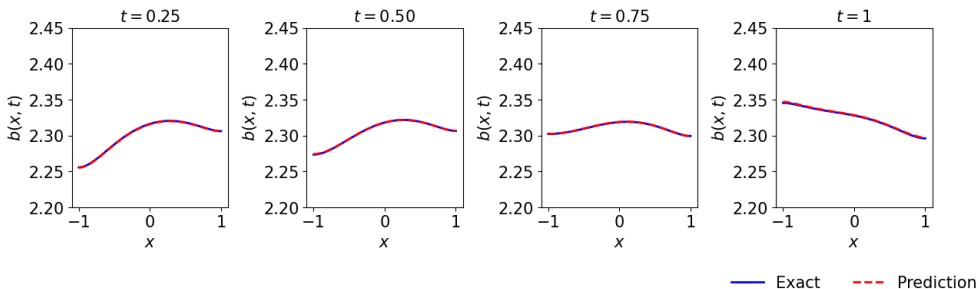

Figure 5: Calbindin concentration along neuron over time

## 5 CONCLUSIONS

In this study, we presented a physics-informed neural network (PINN) for modeling a coupled calcium model in a cable neuron. This is significant for the study of repetitive transcranial magnetic stimulation (rTMS), as it allows for long-time simulations with near instantaneous results. While traditional numerical methods provide accurate solutions, the requirement of small timesteps results in significant computation time and hampering the use for simulating a full rTMS treatment that could last several minutes with voltage pulses in millisecond bursts (Shirinpour et al., 2021). Training a PINN on a short timespan allows for feeding data into the fully trained network and producing a long simulation.

Our results show that PINNs are capable of accurately capturing the calcium concentration over a cable neuron, which can be extended over time. While this work is limited to a single cable neuron, using this to create branches, and therefore a full neuron model, is an important direction for future work.

### 5.1 BROADER IMPACT

Transcranial Magnetic Stimulation is an important noninvasive treatment for a wide array of neurological disorders. While the neuronal response to TMS is still poorly understood, simulation tools are providing a way of studying these effects and how they will benefit patients. A limiting factor in this research has been the difficulty in developing long-time simulations using traditional numerical methods. Neural networks provide the ability to provide long-time simulations with ease that may be applied in both a research and clinical setting, benefitting researchers and patients alike. Our work is an important step in that direction, and has the potential to fill this need.

## ACKNOWLEDGMENTS

This work was supported by NIH grant R01EB034143.

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

APPENDIX A: MEMBRANE FLUX COMPONENTS

The plasma membrane flux in this study can be separated as

$$J_{PM} = -J_P - J_N + J_{SYN} + J_{VDCC}. \tag{14}$$

In this section, we give a brief description of each component as described in Borole et al. (2023).

**PMCA Pumps** A second-order Hill equation is used to model the plasma membrane $Ca^{2+}$ current of PMCA pumps,

$$J_P(c_c) = \rho_P \cdot \frac{I_P c_c^2}{K_P^2 + c_c^2}, \tag{15}$$

where $\rho_P$ is the density of PMCA pumps on the plasma membrane, $I_P$ is the single channel $Ca^{2+}$ current, $c_c$ is the cytosolic $Ca^{2+}$ concentration, and $K_P$ is the measure of $Ca^{2+}$ affinity.

**NCX Exchangers** A second-order Hill equation is used to model the plasma membrane $Ca^{2+}$ current of NCX exchangers,

$$J_N(c_c) = \rho_N \cdot \frac{I_N c_c}{K_N + c_c} \tag{16}$$

where $\rho_N$ is the density of NCX exchangers on the plasma membrane, $I_N$ is the single channel $Ca^{2+}$ current, $c_c$ is the cytosolic $Ca^{2+}$ concentration, and $K_N$ is the measure of $Ca^{2+}$ affinity.

**Synaptic Influx**

$$J_{SYN} = j_{rls} \cdot \left(1 - \frac{t - t_0}{\tau_{rls}}\right) \lambda_{t_0}(t) \tag{17}$$

$$\lambda_{t_0}(t) = \begin{cases} 1, & t \in [t_0, t_0 + \tau_{rls}] \\ 0, & \text{otherwise.} \end{cases} \tag{18}$$

**VDCCs** For VDCCs, we use a Borg-Graham model. The $Ca^{2+}$ current is given by

$$J_{VDCC}(V, c_c, t) = G(V, t)F(V, \Delta[Ca^{2+}]), \tag{19}$$

where $G(V, t) \in [0, 1]$ is the gating function and $F(V, \Delta[Ca^{2+}])$ is the flux function. The difference between cytoplasmic and extracellular ion concentration is given by

$$\Delta[Ca^{2+}] = c_c - c_o,$$

and the flux function is given by the Goldman-Hodgkin-Katz equation,

$$F(V, \Delta[Ca^{2+}]) = \overline{p}_{Ca^{2+}} \frac{V z^2 F^2}{RT} \cdot \frac{c_c - c_o \exp(-zFV/RT)}{1 - \exp(zFV/RT)} \tag{20}$$

where $R$ is the universal gas constant, $F$ is Faraday's constant, $T$ is temperature in Kelvin, $\overline{p}_{Ca^{2+}}$ is the permeability of $Ca^{2+}$ ions through the channels, and $z$ is the valence of a $Ca^{2+}$ ion. The gating function for N-type VDCCs is given by

$$G(V, t) = k(V, t)l^2(V, t), \tag{21}$$

where the gating functions $k(\cdot)$ and $l(\cdot)$ satisfy the two ODEs

$$\frac{\partial k}{\partial t} = \frac{k_\infty - k}{\tau_k} \quad \text{and} \quad \frac{\partial l}{\partial t} = \frac{l_\infty - l}{\tau_l},$$

where

$$k_\infty = \frac{\alpha_k(V)}{\alpha_k(V) + \beta_k(V)}; \quad l_\infty = \frac{\alpha_l(V)}{\alpha_l(V) + \beta_l(V)},$$

and

$$\tau_k = \frac{1}{\alpha_k + \beta_k} + \tau_{k,0}; \quad \tau_l = \frac{1}{\alpha_l + \beta_l} + \tau_{l,0}.$$

The rate functions are defined as

$$\alpha_k(V) = K_k \exp\left(\frac{z_k \gamma_k (V - V_{1/2,k})F}{RT}\right)$$

$$\beta_k(V) = K_k \exp\left(\frac{-z_k(1 - \gamma_k)(V - V_{1/2,k})F}{RT}\right)$$

$$\alpha_l(V) = K_l \exp\left(\frac{z_l \gamma_l (V - V_{1/2,l})F}{RT}\right)$$

$$\beta_l(V) = K_l \exp\left(\frac{-z_l(1 - \gamma_l)(V - V_{1/2,l})F}{RT}\right)$$

APPENDIX B: CONSTANT VALUES

Table 1: Values of constants in calcium model

| Constant | Value |
|---|---|
| $D$ | 1 |
| $b^{tot}$ | 5 |
| $k_b^-$ | 6 |
| $k_b^+$ | 3 |
| $J_P$ | |
| $\rho_P$ | 2 |
| $I_P$ | 1E-2 |
| $K_P$ | 3 |
| $J_N$ | |
| $\rho_N$ | 2E5 |
| $I_N$ | 1E-4 |
| $K_N$ | 180 |
| $J_{SYN}$ | |
| $j_{rls}$ | 1 |
| $J_{VDCC}$ | |
| $R$ | 8.314 |
| $F$ | 96485 |
| $T$ | 310 |
| $z$ | 2 |
| $\tau_{k,0}$ | 1.7E-3 |
| $\tau_{l,0}$ | 70E-3 |
| $K_k$ | 1.7E-3 |
| $K_l$ | 70E-3 |
| $z_k$ | 2 |
| $z_l$ | 1 |
| $\gamma_k$ | 0 |
| $\gamma_l$ | 0 |
| $V_{1/2,k}$ | -21E-3 |
| $V_{1/2,l}$ | -40E-3 |

