# OpenReview forum: "A PHYSICS-INFORMED NEURAL NETWORK FOR COUPLED CALCIUM DYNAMICS IN A CABLE NEURON"
_ICLR.cc/2024/Workshop/AI4DiffEqtnsInSci — AI4DiffEqtnsInSci @ ICLR 2024 Poster_

### Official Review · Reviewer_Hj9J · 2024-02-23
**An exploration on applying PINNs to predicting calcium dynamics**

**Rating:** 6
**Confidence:** 3

**Review:**

This work details a PINN for modeling calcium dynamics, with application to Transcranial Magnetic Stimulation (TMS). The work is exploratory in nature and mentions that the goal is to extend the work to full neuron model and to simulations that extend over several minutes.

I know little about TMS, but I can appreciate the effort it appears to have taken to construct this PINN.

Some experimental details are unclear, particularly with respect to the training/val/test data splits. It looks like the PINN is trained and tested on the same temporal range `t=[0,1]`, and I would have expected to see some results where it was trained on, say, `t=[0,0.8)` and tested on `t=[0.80,1]`. The results in Figure 2 and Figure 3 look great! However, the current test split likely isn't challenging enough to inform about how well this model will generalize in practice. An ablation study to quantify the improvement gained by using the `sin` activation function instead of `tanh` is missing as well.

---

### Official Review · Reviewer_ZAR9 · 2024-02-27
**Are PINN's really needed for the application?**

**Rating:** 6
**Confidence:** 2

**Review:**

The paper proposes the use of a PINN to model calcium dynamics in (biological neurons). This is motivated by the high computational costs of simulating this using conventional methods which prevents long-term simulations. However, since the proposed PINN has 4 x 50 = 200 parameters, I am quite surprised that this is infeasible to simulate conventionally (also, it's a 1D PDE and it's not clear why/when it is stiff). The numerical results show very good agreement between conventional simulation and PINN, but I again wondering why a (artificial) neural network is needed when high accuracy can be achieved with such a simple neural network.

---

### Meta-Review · Area_Chair_qyfm · 2024-03-01

**Recommendation:** Accept (Poster)

**Metareview:**

The paper proposes a PINN based framework for magnetic simulation of calcium dynamics. The reviewers pointed out a few important things or questions which would have been great that authors could address it. That makes me be at the reject/accept border about this work. Due to their effort on using AI for this problem and under the condition that they address the reviewers' questions and also clarify a few parts mentioned in the review, I will vote for accept (poster).

---

### Decision · Program_Chairs · 2024-03-01

Accept (Poster)